# Value-State Gated Attention for Mitigating Extreme-Token Phenomena in Transformers

## Abstract

Large models based on the Transformer architecture are susceptible to extreme-token phenomena, such as attention sinks and value-state drains. These issues, which degrade model performance, quantization fidelity, and interpretability, arise from a problematic mutual reinforcement mechanism where the model learns an inefficient 'no-op' behavior by focusing attention on tokens with near-zero value states. In this paper, we propose Value-State Gated Attention (VGA), a simple, dedicated, and stable architectural mechanism for performing 'no-op' attention efficiently by directly breaking this cycle. VGA introduces a learnable, data-dependent gate, computed directly from the value vectors (V), to modulate the output. Through a theoretical analysis of the underlying gradients, we show that gating the value-state with a function of itself is more effective at decoupling value and attention score updates than prior methods that gate on input embeddings. This creates a direct regulatory pathway that allows the model to suppress a token's contribution based on its emergent value representation. Our experiments demonstrate that VGA significantly mitigates the formation of attention sinks and stabilizes value-state norms, leading to improved performance, robust quantization fidelity, and enhanced model interpretability.

## 1 Introduction

The Transformer architecture has revolutionized artificial intelligence in various domains (Achiam et al., 2023; Dubey et al., 2024; Guo et al., 2025; Dosovitskiy et al., 2020; Brooks et al., 2024). Much of this success stems from the attention mechanism, which allows for efficient and scalable modeling of long-range dependencies. Despite their remarkable success, these models are susceptible to a set of persistent and detrimental emergent behaviors collectively known as **extreme-token phenomena** (Xiao et al., 2023b; Guo et al., 2024; He et al., 2024; Hu et al., 2024). These manifest as a trio of interconnected pathologies: attention sinks (Bondarenko et al., 2023; Guo et al., 2024; Sun et al., 2024; Gu et al., 2024; Barbero et al., 2025; Qiu et al., 2025; Agarwal et al., 2025; Kang et al., 2025), the tendency for certain tokens to receive disproportionately high attention weights regardless of semantic relevance; value-state drains (Guo et al., 2024; Zhou et al., 2024), where the value vectors of these sink tokens exhibit pathologically small norms; and residual-state peaks (Sun et al., 2024; Guo et al., 2024), the abnormal growth of the residual-state norms of the sink token in deeper models.

These interconnected pathologies have profound negative impacts on model performance, quantization fidelity, and interpretability. The pathological growth of residual-state norms induces numerical instability, causing divergence or stagnant learning during training (Zhai et al., 2023). Furthermore, the extreme dynamic range of activations created by these phenomena poses significant challenges for model quantization, often resulting in a substantial accuracy degradation (Bondarenko et al., 2023; Su & Yuan, 2025). Crucially, attention sinks also compromise model interpretability (Bondarenko et al., 2023; Darcet et al., 2023; Wang et al., 2024; Lappe & Giese, 2025), as attention weights no longer reliably indicate semantic importance. The systemic nature of these issues across various Transformer models underscores the need for a fundamental architectural solution.

Recent analyses suggest that these phenomena are not isolated failures, but symptoms of a single underlying pathological feedback loop known as the **mutual reinforcement cycle** (Guo et al., 2024). This cycle arises from the interplay between the optimization dynamics and the competitive nature of the softmax function, which constrains the attention weights to sum to one. The cycle begins when

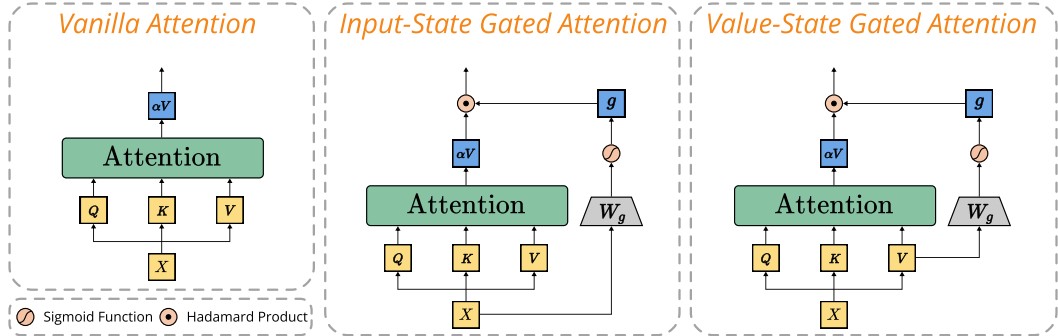

Figure 1: Architecture of Value-State Gated Attention (VGA). Unlike vanilla attention or input-state gated attention, VGA introduces a value-state gating mechanism to modulate the attention output.

an attention head, lacking a relevant context token but forced to distribute its entire attention budget, directs it to a structurally convenient token to perform a 'no-op' behavior (Bondarenko et al., 2023; He et al., 2024; Hu et al., 2024), thereby creating an attention sink. To prevent this arbitrarily high attention from corrupting the output, the optimizer reactively minimizes the norm of the sink token's value vector, leading to a drain of the value state. This drained value state then makes the token an even more attractive target for future 'no-op' attention, reinforcing the cycle and locking the model into a stable but pathological equilibrium. The mutual reinforcement cycle is a classic example of an *unstable positive feedback loop* in control theory (Ogata, 2010).

From the analysis of the mutual reinforcement cycle, we observe that the value state is highly indicative of the sink token, which can be leveraged to directly break the mutual reinforcement cycle. Based on this observation, we propose **V**alue-State **G**ated **A**ttention (**VGA**, as illustrated in Figure 1), a novel and efficient mechanism that introduces a learnable *negative feedback controller* that stabilizes the optimization dynamics, and enables the model to perform a 'no-op' without resorting to attention sinks and value destructions, thus mitigating extreme-token phenomena.

More specifically, we introduce a learnable, value-state dependent gate to modulate the output of the attention head. The core of our approach lies in making the gate's behavior a reactive function of the value state itself, establishing a direct regulatory pathway that allows the model to suppress a token's contribution based on its emergent value representation. This design establishes a feedback control system for mitigating extreme-token phenomena. When the optimization process begins to induce a value-state drain, our gate, being directly dependent on this state, learns to close. By closing, the gate severs the gradient flow that drives the value drains. By decoupling high attention from the pressure to suppress value norms, VGA breaks the mutual reinforcement cycle at its core, thereby preventing the formation of extreme-token phenomena.

In this paper, we introduce VGA—a simple dedicated and stable architectural mechanism for efficient performing of 'no-op' attention, resolving an inherent optimization conflict in the standard attention, and provide a comprehensive analysis of its theoretical underpinnings. Our empirical validation demonstrates that VGA significantly mitigates the formation of attention sinks and stabilizes value-state norms, leading to improved performance, robust quantization fidelity, and enhanced model interpretability. VGA is intentionally designed to be orthogonal to the attention score computation. By functioning as a lightweight, additive gate on the output, it leaves the capabilities of the attention mechanism intact. This architectural separation makes VGA a general and minimally invasive enhancement, readily applicable to any Transformer-based model.

## 2 RELATED WORK

**Extreme-token phenomena and understanding.** Research on Transformers has identified a set of issues, collectively termed extreme-token phenomena. Initial work identified the phenomenon of attention sinks, where certain tokens attract a disproportionate amount of attention (Xiao et al., 2023b; Darcet et al., 2023; Guo et al., 2024; Gu et al., 2024; Barbero et al., 2025; Qiu et al., 2025; Agarwal et al., 2025; Kang et al., 2025). Subsequent work linked this observation to value-state drains, noting that these same sink tokens consistently have value states with unusually small norms (Guo et al., 2024; Zhou et al., 2024). A mechanistic explanation for this connection was a

mutual reinforcement cycle in the training process. This cycle arises when an attention head performs a no-op operation (Bondarenko et al., 2023; He et al., 2024; Hu et al., 2024), triggering an irreversible loop between high attention weights and the optimizer's suppression of the corresponding value-state norms. Further research in deeper models also documented the emergence of residual-state peaks, an abnormal growth in the residual states of sink tokens (Sun et al., 2024; Guo et al., 2024).

**Mitigating extreme-token phenomena in post-processing.** Existing methods could be categorized into weight-modifying approaches that often involve lightweight fine-tuning (Wang et al., 2024; Chen et al., 2024), and state-manipulation approaches that operate intermediate states at inference time without altering model weights (Son et al., 2024; Yu et al., 2024; Li et al., 2023). A noteworthy finding in Xiao et al. (2024) is the necessity of deliberately reintroducing the "attention sinks" introduced from the pre-training phase into downstream tasks to maintain the model's performance. While remedial solutions in the post-processing stage can alleviate problems in the downstream usage of pretrained models, a more fundamental approach is to enhance the models' intrinsic capabilities and prevent these issues from arising during the pretraining phase.

**Mitigating extreme-token phenomena in pretraining.** Several strategies have been proposed to mitigate extreme-token phenomena during pretraining. One line of work introduces special tokens that are designed to absorb unneeded attention, such as register tokens (Darcet et al., 2023; Barbero et al., 2025; Lappe & Giese, 2025). Another group of researchers modifies the attention mechanism itself to prevent or control attention concentration. These proposals include replacing softmax with alternative functions (Ramapuram et al., 2024; Gu et al., 2024), adding a clipping function to attention scores (Bondarenko et al., 2023), or creating a learnable sink within the mechanism to channel attention (Xiao et al., 2023b; Miller, 2023; Agarwal et al., 2025). A third approach uses gating to control a token's contribution. While gating has been widely used in LSTMs (Hochreiter & Schmidhuber, 1997) and GRUs (Dey & Salem, 2017), as well as in Transformers(*e.g.*, Yang et al. (2024); Gao et al. (2025)), its application to resolve optimization pathologies remains less explored. Within this domain, prior approaches (Bondarenko et al., 2023; Qiu et al., 2025) have focused on learning the predictive gating function from the input embeddings. While prior methods have focused on providing static sinks for extraneous attention or predictive gates based on input features, our work is the first to propose a fully reactive, self-regulatory gate that directly addresses the emergent optimization dynamics of the value-state.

A primary motivation for resolving these phenomena is to improve numerical stability of a model, particularly for Post-Training Quantization (PTQ). Extreme-token phenomena create large activation outliers, which significantly increases the dynamic range of tensors. This poses a major challenge for PTQ methods (Li et al., 2021; Xiao et al., 2023a), as mapping a wide value range to a low-bit format can cause substantial precision loss and performance degradation. As noted by Bondarenko et al. (2023), models exhibiting extreme-token behaviors are not quantization-friendly. Therefore, mitigation strategies applied during pretraining are expected to result in models that are easier to quantize. Following this line of reasoning, we adopt a common PTQ methodology (Bondarenko et al., 2023) in our experiments to demonstrate that successfully mitigating extreme-token phenomena leads to improved post-training quantization results.

## 3 METHOD

This section details the theoretical mechanisms of extreme-token phenomena and introduces our proposed solution, Value-State Gated Attention. In Section 3.1, we provide a formal gradient-based analysis of the mutual reinforcement cycle, first identified by Guo et al. (2024), in the training of the standard attention module. We also discuss the pioneering Input-State Gated Attention (IGA) approach (Bondarenko et al., 2023; Qiu et al., 2025) for mitigating the extreme-token phenomena. Our analysis reveals the limitations of IGA, highlighting that the problem is fundamentally tied to the optimization dynamics of the value-state itself, but not the input-state. Building on this insight, Section 3.2 details the architecture of VGA, and Section 3.3 provides a theoretical analysis demonstrating how its design inherently breaks this cycle by altering the gradient dynamics.

### 3.1 PRELIMINARIES

For clarity, our discussion focuses on the vanilla single-head self-attention, although the principle we introduce is readily applicable to more complex variants. The mechanism computes a representation for each token by attending to all tokens in the input sequence. Given an input sequence $X \in \mathbb{R}^{n \times d}$,

it is projected into Query ($Q$), Key ($K$), and Value ($V$) matrices via learnable weight matrices $W_Q, W_K, W_V \in \mathbb{R}^{d \times d}$, such that $Q = XW_Q$, $K = XW_K$, and $V = XW_V$. The attention weight matrix $\alpha$ is then computed as:

$$\alpha = \text{softmax}\left(\frac{QK^T}{\sqrt{d}}\right). \tag{1}$$

The final attention output is computed as $\text{Attention}(Q, K, V) = \alpha V$. The matrix $\alpha \in \mathbb{R}^{n \times n}$ contains the attention weights, with each element $\alpha_{ij}$ representing the weight that query $i$ assigns to key $j$. The softmax function ensures that the weights for each query are non-negative and sum to one, *i.e.*, $\forall i, \sum_{j=1}^{n} \alpha_{ij} = 1$. Consequently, the output for the $i$-th token $z_i$ is a convex combination of all value vectors in the sequence:

$$z_i = \sum_{j=1}^{n} \alpha_{ij} V_j. \tag{2}$$

This formulation becomes problematic when an attention head needs to perform a 'no-op' (*i.e.*, contribute minimally to the output). Due to the softmax constraint, the head is forced to distribute its attention budget across tokens. If no token is semantically relevant, the head may learn to dump its attention onto a single, structurally convenient sink token. Guo et al. (2024) first identified this underlying issue as the mutual reinforcement cycle: a pathological feedback dynamic that drives the formation of attention sinks and the accompanying suppression of their value states (value-state drains). To formalize this process, we analyze the cycle from the perspective of gradient dynamics.

To understand the optimizer's behavior, we examine the gradient of the loss $L$ with respect to an arbitrary value vector $V_j$. Applying the chain rule to Equation (2), we obtain:

$$\frac{\partial L}{\partial V_j} = \sum_{i=1}^{n} \frac{\partial z_i}{\partial V_j} \frac{\partial L}{\partial z_i} = \sum_{i=1}^{n} \alpha_{ij} \frac{\partial L}{\partial z_i}, \tag{3}$$

where $\frac{\partial L}{\partial z_i}$ represents the upstream gradient. This reveals a rigid coupling: the gradient flowing to $V_j$ is a direct scaling of the upstream gradients by the attention weights $\alpha_{ij}$.

The severity of this coupling becomes apparent in the limiting case where a token $s$ becomes a perfect attention sink for a query $i$, *i.e.*, $\alpha_{is} \to 1$. The gradient flowing to its value vector $V_s$ from this query approaches the full upstream gradient $\frac{\partial L}{\partial z_i}$, while all other value vectors $V_j$ ($j \neq s$) receive a vanishing gradient. To perform a 'no-op' and minimize the output's magnitude despite a large attention weight $\alpha_{is}$, the amplified gradient provides a strong signal to reduce the magnitude of $V_s$. This forces the optimizer to aggressively push the norm of $V_s$ towards zero, precipitating a value-state drain. As illustrated in Figure 2, this initiates *an unstable positive feedback loop*: high attention amplifies the gradient to $V_s$, the optimizer suppresses its norm, and the resulting inert value state becomes an even safer target for future no-op queries.

A prominent prior approach, Input-State Gated Attention (IGA) approach (Bondarenko et al., 2023; Qiu et al., 2025) is proposed to address the problem, which introduces a gate to control information flow. The gate $g_j$ is computed from the input embeddings $X_j$ via a learnable weight matrix $W_g \in \mathbb{R}^{d \times 1}$, followed by a sigmoid activation function $\sigma$, as $g_j = \sigma(X_j W_g)$, modulating the attention output as:

$$z_i = \sum_{j=1}^{n} g_j \alpha_{ij} V_j. \tag{4}$$

This architecture introduces a crucial second weighting path. The vanilla attention scores $\alpha$ continue to determine where the model attends, while the new input-state gates $g$ independently determine how much information is received. The model learns to predict from $X_j$ whether token $j$ is a 'no-op' candidate and closes the gate $g_j$. To analyze its effect on the value-state drains, we examine the gradient with respect to $V_j$. Since $g_j$ is a function of $X_j$, it is treated as a constant with respect to $V_j$:

$$\frac{\partial L}{\partial V_j} = \sum_{i=1}^{n} g_j \alpha_{ij} \frac{\partial L}{\partial z_i}. \tag{5}$$

When a token becomes an attention sink where $\alpha_{is} \to 1$, the model would learn to close the corresponding gate, $g_s \to 0$, to nullify the contribution of the sink token.

However, the fundamental coupling remains. The optimizer still perceives a direct path to nullify the output by shrinking $V$, as the gate $g$ is based upon $X$, not the emergent dynamics of $V$. Hence, the gating control is predictive and indirect.

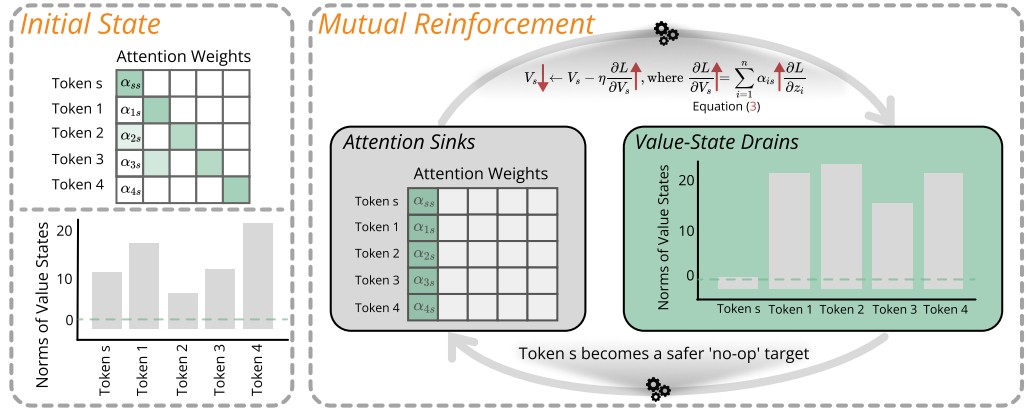

Figure 2: The mutual reinforcement cycle that leads to attention sinks and value-state drains. (Left) Initial state with natural attention weights and moderate value state norms. (Right) The cycle begins when a query allocates high attention to a sink token $s$. This amplifies the gradient backpropagated to $V_s$, prompting the optimizer to suppress its norm via learning rate $\eta$, resulting in a value-state drain. This suppression makes the token an even safer target for future 'no-op' queries, locking it into the sink role.

## 3.2 VALUE-STATE GATED ATTENTION

This analysis of IGA reveals the problem's essence: the instability originates from the optimization pressure applied directly to the value state itself. Therefore, an effective solution cannot be merely predictive; it must be reactive, creating a control directly from the value state. This insight motivates our proposed architecture: Value-State Gated Attention (VGA). As illustrated in Figure 1 (Right), VGA operates as a reactive control system. It introduces a learnable gate computed directly from the value vectors, which modulates each token's contribution to the attention output. This design endows the model with a direct mechanism to perform a 'no-op' by selectively attenuating information flow based on a token's current value state.

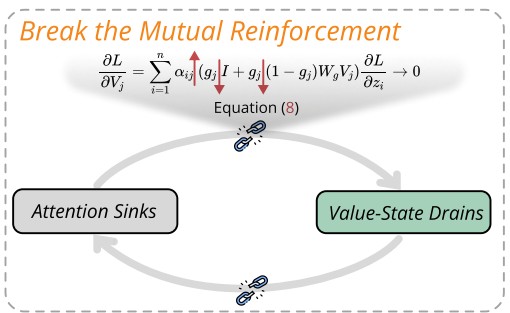

Figure 3: VGA alters gradient dynamics by learning to close the gate for an attention sink ($g_s \rightarrow 0$). This action severs the gradient flow to the value state $V_s$, effectively breaking the cycle.

Formally, VGA introduces a gating vector $g_j$ for each value vector $V_j$. This is achieved via a linear projection with a learnable weight matrix $W_g \in \mathbb{R}^{d \times 1}$, followed by a sigmoid activation function $\sigma$:

$$g_j = \sigma(V_j W_g). \tag{6}$$

These gates are then applied to the output for an individual token $i$ same as that in Equation (4). A more detailed description of VGA is provided in Appendix B.

The ONE AND ONLY critical distinction between IGA and VGA lies in the basis for the gating function: the input-state versus the value-state. Consequently, VGA retains the "second weighting path" advantage of IGA. Moreover, its computational and parameter overhead is identical to IGA, and remains marginal (one projection to low-dimensions and element-wise operations) relative to the total amount of the vanilla model.

## 3.3 HOW VGA BREAKS THE MUTUAL REINFORCEMENT

We now demonstrate how the minimal change of gating basis from input-state to value-state acts as *a learnable negative feedback controller* that stabilizes the optimization dynamics and breaks the mutual reinforcement cycle by dramatically altering the gradient landscape. As shown in Figure 3, VGA decouples attention magnitude from gradient flow through a reactive control mechanism.

**Gradient dynamics in VGA.** In VGA, a value vector $V_j$ influences the output in two ways: it provides the semantic content, and it also determines its own transmission strength via the gate computation. This necessitates applying the product rule when deriving the gradient of the loss $L$ with respect to $V_j$. Considering the contribution from a single query $i$ for clarity, the gradient is:

$$\frac{\partial L}{\partial V_j} = \sum_{i=1}^{n} \alpha_{ij} \frac{\partial (g_j V_j)}{\partial V_j} \frac{\partial L}{\partial z_i}. \tag{7}$$

The derivative of the gated value expands into two components corresponding to the two influence paths:

$$\frac{\partial L}{\partial V_j} = \sum_{i=1}^{n} \alpha_{ij} \left( \underbrace{g_j I}_{\text{Content Path}} + \underbrace{g_j(1 - g_j)W_g V_j}_{\text{Self-regulatory Path}} \right) \frac{\partial L}{\partial z_i}, \tag{8}$$

where the 'Content Path' term, $g_j I$, reflects the gradient through the vector's role as content, modulated by its gate $g_j$. The 'Self-regulatory Path' arises from the self-referential design, where $V_j$ influences its own gate. This formulation is the key to VGA's efficacy. We analyze its behavior in the critical scenario where a token $s$ is an attention sink ($\alpha_{is} \to 1$) and the model has learned to close its gate ($g_s \to 0$). In this limit, both gradient paths are nullified: (1) The 'Content Path' term $g_s I$ vanishes as $g_s \to 0$; (2) The 'Self-regulatory Path' term also vanishes, as it contains both $g_s$ and the sigmoid derivative factor $g_s(1 - g_s)$, which also approaches zero. Consequently, the gradient flowing to $V_s$ is entirely severed: $\frac{\partial L}{\partial V_s} \to 0$, as $g_s \to 0$. This demonstrates that VGA successfully decouples the attention weight from the gradient magnitude. Even when attention is maximal, closing the gate provides a clean 'no-op' mechanism that breaks the link between high attention and the pathological gradient amplification, thus dismantling the mutual reinforcement cycle. This analysis demonstrates that VGA provides a direct gradient pathway to perform a 'no-op' without value-state suppression. Our empirical validation in Section 4.1 confirms that the optimizer successfully learns to utilize this mechanism, preferring to close the gate rather than pathologically shrinking the value norms.

Also note that the term $g_j(1 - g_j)$ in the 'Self-regulatory Path' is maximal when the gate is in its most uncertain state ($g_j = 0.5$) and diminishes as the gate becomes confident in its decision (approaching 0 or 1). This ensures that the self-regulatory feedback is strongest when it is most needed—during the transition—and weakest when the gate is already in a stable open or closed state. This property prevents runaway feedback loops and is a hallmark of a well-designed, stable control system.

## 4 EXPERIMENTS

Extensive experiments and comparisons are conducted to demonstrate the advantage of VGA in various settings. We begin in Section 4.1 by validating VGA on a specifically designed synthetic task as proposed by Guo et al. (2024). Then we evaluate VGA on several language models in Section 4.2, including BERT and OPT—following the practice of Bondarenko et al. (2023), as well as GPT-2—a seminal autoregressive model whose design became the foundation for numerous subsequent large-scale commercial models, and the evidence from which is highly instructive for training large-scale commercial LLMs (Allen-Zhu, 2024). Finally, we show the quantization results of BERT and OPT models (again, following the practice of Bondarenko et al. (2023)) with VGA in Section 4.3.

### 4.1 EMPIRICAL VALIDATION ON THE BIGRAM-BACKCOPY TASK

To empirically validate our analysis, we use the Bigram-Backcopy (BB) task from Guo et al. (2024). This synthetic task is specifically designed to create a controlled environment where extreme-token phenomena are reliably induced. Data generation follows two rules based on token identity: 1) **Bigram task:** For most non-trigger tokens, the next token is generated from a fixed bigram probability distribution, similar to a Markov chain. The attention mechanism provides no useful information for this task. 2) **Backcopy task:** When predefined trigger tokens (*e.g.*, t) is encountered, the model must ignore the bigram rule and instead copy the token preceding the trigger. This task requires the attention mechanism to focus specifically on the token to be copied. We trained two models on this task: a vanilla Transformer and one with VGA. The results are presented in Figure 4 and Figure 5.

**Attention sinks.** Figure 4(a) shows that the two models exhibit clearly distinct attention patterns. The vanilla model (left) exhibits a prototypical attention sink. For all non-trigger query tokens, attention

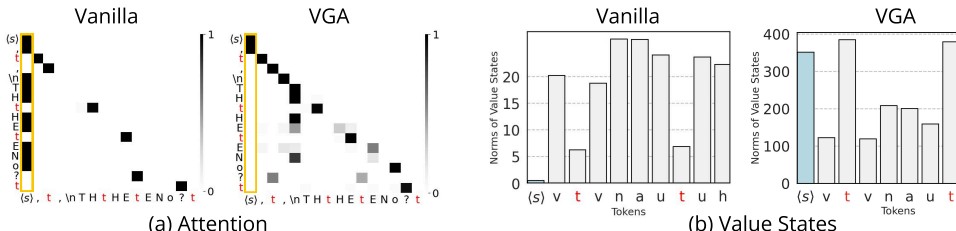

(a) Attention             (b) Value States

Figure 4: Comparative analysis of a vanilla Transformer (left) and a VGA model (right) on the Bigram-Backcopy (Guo et al., 2024) task. (a) VGA prevents the formation of an attention sink on the  token. (b) Consequently, VGA resolves the corresponding value-state drain, preserving the norm of the sink token's value vector.

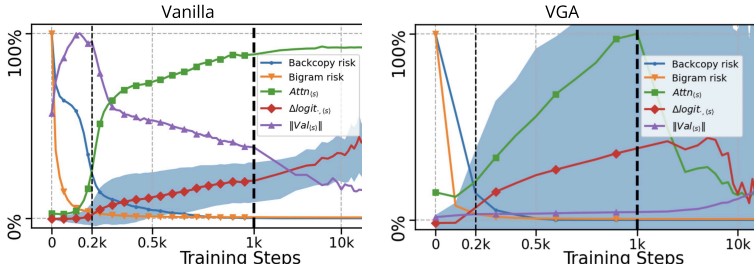

Figure 5: Training dynamics of a vanilla Transformer (left) versus a VGA model (right) on the Bigram-Backcopy task, tracking task performance and sink-token metrics over training steps.

collapses onto the  (start) token, which carries no semantic information for these queries. In contrast, VGA (right) mitigates this behavior. The attention weights are more evenly distributed, and no single token acts as a universal sink, demonstrating that VGA effectively prevents the pathological concentration of attention.

**Value-state drains.** Mitigating attention sinks directly corresponds to the stabilization of value-state norms, as illustrated in Figure 4(b). In the vanilla model (left), the  token suffers from a severe value-state drain, with its norm aggressively reduced toward zero. This reflects the optimizer's response to the high attention weights, consistent with theoretical analysis. In the VGA model (right), this pathology is absent. The norm of the  token's value state is maintained at a healthy, non-pathological level comparable to other tokens in the sequence, confirming that our mechanism removes the pressure to destroy value representations.

**Training dynamics.** The evolution of key metrics during training, presented in Figure 5, provides clear evidence that VGA breaks the mutual reinforcement cycle. We track several metrics: (i) Backcopy risk and Bigram risk measure model performance on the two sub-tasks, with lower values being better. (ii) $Attn_{}$ is the average attention weight directed to the  token from non-trigger queries. (iii) $||Val_{}||$ is the L2 norm of the  token's value state. (iv) $\Delta logit._{}$ is calculated by taking the attention logit for the sink key  and subtracting the mean of the logits for all other keys, averaged over non-trigger queries. A higher $\Delta logit._{}$ indicates a stronger pre-softmax bias toward the sink token. The vanilla model (left) exhibits the cycle's characteristic signature: as the model learns, $Attn_{}$ and $\Delta logit._{}$ steadily increase. Concurrently, $||Val_{}||$ collapses toward zero. This inverse relationship is the hallmark of the pathological feedback loop. The dynamics of the VGA model (right) present a fundamentally different result. While the model learns the task equally well, the pathological signals all remain stable at healthy, non-extreme levels throughout training.

**Enhanced interpretability.** VGA enhances interpretability by using a gate $g_j$ to disentangle a token's attention weight from its informational contribution. This provides an explicit signal for a 'no-op' operation, where high attention can be paired with a near-zero gate. This resolves a key ambiguity of vanilla attention, where a high attention score can signify either importance or pathology. The results in Figure 4 validate this enhancement by illustrating the vanilla model's ambiguous state, where high attention on the sink token is paired with the suppression of its value norm. The VGA model exhibits stable attention and value norms, demonstrating its dedicated no-op mechanism prevents the formation of the pathological attention sinks. Consequently, VGA's attention patterns become a more faithful and reliable indicator of the model's internal information seeking process.

Table 1: Comparison of our proposed VGA against several baselines on BERT (Devlin et al., 2019), OPT-125m (Zhang et al., 2022), and GPT-2 (Radford et al., 2019) models. We evaluate task performance using Perplexity and model stability by measuring the Max I+O Norm and Avg. kurtosis to quantify activation outliers. The best result for each metric is marked in bold. VGA consistently delivers top-tier perplexity while suppressing extreme values, demonstrating its dual benefit of boosting performance and improving model stability.

| Model | Method | Perplexity ($\downarrow$) | Max I+O Norm ($\downarrow$) | Avg. kurtosis ($\downarrow$) |
|---|---|---|---|---|
| BERT | Vanilla | $4.52^{\pm0.03}$ | $812.39^{\pm141.62}$ | $2987.23^{\pm313.21}$ |
| | Register Tokens | $4.53^{\pm0.01}$ | $1205.91^{\pm223.14}$ | $7812.73^{\pm1281.12}$ |
| | Learnable Sink | $4.53^{\pm0.02}$ | $41.65^{\pm7.83}$ | $225.88^{\pm41.81}$ |
| | IGA | $4.53^{\pm0.01}$ | $38.71^{\pm10.19}$ | $90.65^{\pm5.78}$ |
| | VGA | $\mathbf{4.52^{\pm0.00}}$ | $\mathbf{33.19^{\pm6.75}}$ | $\mathbf{84.55^{\pm2.84}}$ |
| OPT | Vanilla | $15.95^{\pm0.03}$ | $1.01^{\pm0.04}$ | $2177^{\pm274}$ |
| | Register Tokens | $15.85^{\pm0.04}$ | $0.75^{\pm0.03}$ | $8322.68^{\pm1197.74}$ |
| | Learnable Sink | $15.92^{\pm0.01}$ | $0.77^{\pm0.03}$ | $\mathbf{22.23^{\pm6.77}}$ |
| | IGA | $15.65^{\pm0.01}$ | $0.50^{\pm0.02}$ | $102^{\pm1.32}$ |
| | VGA | $\mathbf{15.49^{\pm0.00}}$ | $\mathbf{0.45^{\pm0.03}}$ | $34.77^{\pm4.65}$ |
| GPT-2 | Vanilla | $17.03^{\pm0.00}$ | $256.84^{\pm18.77}$ | $13412.71^{\pm3,528.30}$ |
| | Register Tokens | $\mathbf{16.26^{\pm0.01}}$ | $73.81^{\pm5.86}$ | $4871.96^{\pm803.12}$ |
| | Learnable Sink | $16.51^{\pm0.00}$ | $14.50^{\pm2.27}$ | $71.08^{\pm6.32}$ |
| | IGA | $16.62^{\pm0.01}$ | $17.68^{\pm4.41}$ | $33.51^{\pm5.29}$ |
| | VGA | $16.52^{\pm0.01}$ | $\mathbf{12.37^{\pm1.28}}$ | $\mathbf{31.27^{\pm6.71}}$ |

Note that the solution proposed by Guo et al. (2024) involves replacing softmax with ReLU, which is a fundamental alteration to the attention mechanism. Although softmax contributes to the mutual reinforcement cycle, it remains central to the model's ability to allocate its finite attention budget. Removing it, as suggested by the use of ReLU, may have unforeseen consequences on model expressivity and behavior. VGA, in contrast, is a lightweight, additive component. It preserves the well-understood properties of the standard softmax-based attention mechanism while surgically correcting a specific failure mode. This makes VGA a more practical and readily adoptable solution.

## 4.2 RESULTS ON REPRESENTATIVE LANGUAGE MODELS

We evaluate our proposed VGA method on several representative transformer-based language models to assess its impact on model performance and activation stability. Experiments follow the framework of Bondarenko et al. (2023) to show that VGA effectively mitigates the extreme-token phenomenon.

**Models and Datasets.** Our evaluation includes three representative language models: BERT (Devlin et al., 2019), OPT-125m (Zhang et al., 2022), and GPT-2 (Radford et al., 2019). Following previous work (Bondarenko et al., 2023), we evaluate BERT and OPT-125m on a combined dataset of BookCorpus (Zhu et al., 2015) and English Wikipedia (Guo et al., 2020). Our GPT-2 implementation follows Karpathy (2022) and incorporates Rotary Position Embeddings (RoPE) (Su et al., 2024), a widely adopted technique in modern Transformer architectures. We train the 124M parameter variant[1] of GPT-2 on the OpenWebText (Gao et al., 2020) dataset.

**Evaluation metrics.** We use Perplexity (Jelinek et al., 1977) to evaluate the overall language modeling performance, where lower values indicate a better predictive capability. To measure activation stability, we adopt two metrics as in Bondarenko et al. (2023). We use the maximum input and output norm (Max I+O Norm) to quantify the magnitude of the most extreme outliers and the average kurtosis (Avg. kurtosis) to measure the heavy-tailedness of the activation distributions. Lower values for both metrics indicate better-behaved distributions, which is crucial for effective quantization (Chmiel et al., 2020; Bondarenko et al., 2021).

**Baselines.** Our chosen baselines are representative of the primary competing paradigms for mitigating extreme-token phenomena, as categorized in our review of related work, allowing for a direct comparison of these distinct approaches. The vanilla softmax attention mechanism serves as our primary benchmark (Vaswani et al., 2017). We also evaluate against Register Tokens (Darcet et al., 2023; Lappe & Giese, 2025), a method that prepends a set of learnable non-semantic tokens as alternative targets for attention heads. Furthermore, we compare with Learnable Sink (Xiao et al., 2023b; Agarwal et al., 2025), which introduces a dedicated token to absorb superfluous attention scores and stabilize attention patterns—a method that serves as a representative for the broader class

---

[1]The training of GPT-2 (124M) requires $8 \times$ A100 40GB for about 5 days. Training larger variants of GPT-2 required resources beyond the scope of this study.

Table 2: Evaluation of post-training quantization on BERT and OPT. $\Delta$Perplexity denotes the perplexity increase relative to the FP32 baseline presented in Table 1.

| Model | Method | INT8 Perplexity ($\downarrow$) | $\Delta$Perplexity (vs. FP32) ($\downarrow$) | Max I+O Norm ($\downarrow$) | Avg. kurtosis ($\downarrow$) |
|---|---|---|---|---|---|
| BERT | Vanilla | $617.32^{\pm 191.20}$ | +612.80 | $486.19^{\pm 227.62}$ | $2508.21^{\pm 1394.71}$ |
| | Register Tokens | $913.67^{\pm 839.10}$ | +909.14 | $415.27^{\pm 218.05}$ | $7,284.18^{\pm 3,365.82}$ |
| | Learnable Sink | $4.79^{\pm 0.03}$ | +0.26 | $42.82^{\pm 2.69}$ | $153.47^{\pm 38.97}$ |
| | IGA | $4.67^{\pm 0.01}$ | +0.15 | $45.06^{\pm 1.50}$ | $91.27^{\pm 2.88}$ |
| | **VGA** | $\mathbf{4.64}^{\pm 0.01}$ | **+0.12** | $\mathbf{37.1}^{\pm 2.37}$ | $\mathbf{78.32}^{\pm 1.82}$ |
| OPT | Vanilla | $43.78^{\pm 6.81}$ | +27.83 | $657.77^{\pm 240.89}$ | $6548.18^{\pm 1567.07}$ |
| | Register Tokens | $123.81^{\pm 91.77}$ | +107.96 | $835.33^{\pm 274.83}$ | $6646.24^{\pm 3735.05}$ |
| | Learnable Sink | $17.31^{\pm 0.01}$ | +1.39 | $114.45^{\pm 4.05}$ | $26.35^{\pm 4.26}$ |
| | IGA | $16.77^{\pm 0.01}$ | +1.12 | $102.15^{\pm 3.81}$ | $95.48^{\pm 6.12}$ |
| | **VGA** | $\mathbf{16.42}^{\pm 0.01}$ | **+0.93** | $100.01^{\pm 3.44}$ | $\mathbf{18.34}^{\pm 1.65}$ |

of strategies on providing an escape from the softmax constraint (Ramapuram et al., 2024; Gu et al., 2024; Bondarenko et al., 2023). Finally, we consider IGA (Bondarenko et al., 2023), representing prior methods that learn the gating function from the input embeddings.

**Results analysis.** The results in Table 1 demonstrate VGA's dual benefit of improving performance while decisively enhancing model stability. Its consistent superiority in reducing Max I+O Norm and Avg. kurtosis across all models indicates a fundamental taming of activation outliers. We observe that other baselines have inherent design limitations. Register Tokens merely provide an alternative sink and can even exacerbate Avg. kurtosis. While Learnable Sink improves stability, its sink mechanism is based on a fixed embedding learned during training, making it incapable of adapting to specific input contexts. In contrast, VGA uses dynamic, data-dependent gating. Similarly, IGA's predictive gating on input embeddings is shown to be less effective at decoupling value-attention updates compared to VGA's reactive mechanism, which operates directly on the emergent value-state. Consequently, VGA provides a more structural solution, one that systemically moderates the entire activation distribution. This makes VGA a robust approach to improving reliability without compromising performance.

### 4.3 POST TRAINING QUANTIZATION RESULTS

We evaluated our method using an 8-bit post-training quantization (PTQ) scheme, following the settings in Bondarenko et al. (2023). As shown in Table 2, the baseline Vanilla models exhibit a large perplexity increase after quantization, corresponding to their high Max I+O Norm and Avg. kurtosis. Notably, Register Tokens results in even greater performance loss. This suggests that simply providing a dedicated sink may concentrate pathological dynamics and exacerbate outlier issues, as evidenced by its extremely high kurtosis. In contrast, all mitigation methods improve quantization robustness, with VGA consistently performing best. On both BERT and OPT, VGA achieves the smallest perplexity increase while also recording the lowest Max I+O Norm and lowest Avg. kurtosis. These results indicate that by addressing the formation of extreme-token phenomena at its source, VGA produces a model with an inherently more quantization-friendly activation distribution. Encouraged by these PTQ results, we plan to incorporate VGA in low-bit pretraining in future work.

## 5 CONCLUSION

In this work, we introduced Value-State Gated Attention (VGA) as a novel and efficient mechanism for mitigating extreme-token phenomena. By computing a learnable gate directly from the value states, VGA creates a self-regulatory mechanism that enables a 'no-op' operation, architecturally decoupling high attention weights from the destructive pressure on value norms. Through both gradient-based theoretical analysis and empirical validation on synthetic tasks and standard models, we demonstrated that VGA effectively mitigates extreme-token phenomena. This directly translates to significant improvements in activation stability and post-training quantization fidelity, often while maintaining or enhancing model perplexity.

As a general design for Transformers, VGA has high potential to be effective in any Transformer-based models, from language models, to vision and multi-modal models, especially when they are scaled up to even larger models, where extreme-token phenomena are even more pronounced. However, such large-scale experiments demand a level of computational resources that places them beyond the scope of the present study. We therefore hope that our work will encourage research groups with access to these resources to investigate VGA's performance in such settings—with only a few lines of code modification. We are optimistic that it will prove to be an impactful component for the next generation of large-scale models across various domains.

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

## A   GRADIENT DERIVATION FOR VALUE-STATE GATED ATTENTION (VGA)

Let $V_j \in \mathbb{R}^{1 \times d}$ be the row vector value for token $j$, $\alpha_{ij}$ the attention weight of query $i$ to token $j$, and $W_g \in \mathbb{R}^{d \times 1}$ the gate projection weight. The gate scalar is computed as:

$$g_j = \sigma(V_j W_g). \tag{9}$$

where $\sigma(\cdot)$ denotes the Sigmoid function.

**VGA output:**

$$z_i = \sum_{j=1}^{n} \alpha_{ij} \left( g_j V_j \right). \tag{10}$$

**Step 1: Base gradient expression.** The gradient of the loss $L$ w.r.t. $V_j$ is:

$$\frac{\partial L}{\partial V_j} = \sum_{i=1}^{n} \alpha_{ij} \frac{\partial (g_j V_j)}{\partial V_j} \frac{\partial L}{\partial z_i}. \tag{11}$$

**Step 2: Product rule for** $(g_j V_j)$. Since $g_j$ depends on $V_j$:

$$\frac{\partial (g_j V_j)}{\partial V_j} = g_j I + \left( \frac{\partial g_j}{\partial V_j} \right)^T V_j, \tag{12}$$

this derivative is written in denominator layout convention, $I \in \mathbb{R}^{d \times d}$ is the identity matrix.

**Step 3: Gate derivative.** Let $u_j = V_j W_g$ (scalar). We have:

$$\frac{\partial u_j}{\partial V_j} = W_g^T, \quad \sigma'(u_j) = g_j(1 - g_j).$$

Therefore:

$$\frac{\partial g_j}{\partial V_j} = g_j(1 - g_j)W_g^T \quad (\in \mathbb{R}^{1 \times d}). \tag{13}$$

**Step 4: Full Jacobian of gated value.** Combining Equation (12) and Equation (13):

$$\frac{\partial (g_j V_j)}{\partial V_j} = g_j I + g_j(1 - g_j)W_g V_j, \tag{14}$$

yielding a $d \times d$ matrix.

**Step 5: Final gradient.** Substituting Equation (14) into Equation (11):

$$\boxed{\frac{\partial L}{\partial V_j} = \sum_{i=1}^{n} \alpha_{ij} \left[ g_j I + g_j(1 - g_j)W_g V_j \right] \frac{\partial L}{\partial z_i}} \tag{15}$$

**Interpretation.** The two terms inside the brackets correspond to two independent gradient pathways:

- *Content Path*: $g_j I$ — gradient through the semantic content of $V_j$, scaled by $g_j$.
- *Self-regulatory Path*: $g_j(1 - g_j)W_g V_j$ — gradient arising from $V_j$ influencing its own gate.

If a sink token $s$ has $\alpha_{is} \to 1$ but $g_s \to 0$, both terms vanish:

$$g_s I \to 0, \quad g_s(1 - g_s) \to 0,$$

cleanly severing the gradient to $V_s$ and breaking the mutual reinforcement cycle.

## B  ARCHITECTURE DETAILS

This section provides a formal and detailed formulation of the Value-State Gated Attention (VGA) mechanism. VGA modifies the standard attention by introducing a gate that modulates the final output of the attention head. This gate, denoted as $g$, is computed directly from the value state $V$, creating a reactive feedback loop.

The core of the VGA mechanism is the computation of a dedicated gate for each attention head. In a multi-head attention setting with $h$ heads and a model dimension of $d$, the gate matrix $g$ is computed for all heads simultaneously from the complete value projection matrix $V \in \mathbb{R}^{N \times d}$. The gate matrix $g \in \mathbb{R}^{N \times h}$ is computed as:

$$g = \sigma(V W_g) \tag{16}$$

where $W_g \in \mathbb{R}^{d \times h}$ is a single learnable weight matrix, and $\sigma(\cdot)$ is the sigmoid function. Each column $g_j$ in the resulting matrix $g$ serves as the specific gate for the $j$-th attention head. The final output for an individual head $j$, denoted $O_{\text{VGA},j}$, is then obtained by an element-wise product between its gate $g_j$ and its standard attention output $O_j$:

$$O_{\text{VGA},j} = g_j \odot O_j \ , \ \text{where } O_j = \text{Attention}(Q_j, K_j, V_j). \tag{17}$$

For this multiplication, the gate vector $g_j \in \mathbb{R}^{N \times 1}$ is broadcast across the feature dimension of the head's output $O_j \in \mathbb{R}^{N \times d_{head}}$, where $d_{head} = d/h$. This allows each head to have its output modulated independently based on the same shared value-state representation. The final layer output is formed by concatenating the gated head outputs, *i.e.*, $\text{Concat}(O_{\text{VGA},1}, \ldots, O_{\text{VGA},h})$, followed by the usual final linear projection.

This design ensures that VGA is a lightweight and minimally invasive module. It introduces only a small number of new parameters $W_g$ per attention head and, crucially, remains orthogonal to the attention score computation. The original attention weights are calculated as usual, preserving the mechanism's ability to model long-range dependencies, while the gate provides a separate, reactive pathway to control the informational output based on the value state.

## C  USING VGA FOR FINE-TUNING PRE-TRAINED MODELS

Table 3: VGA Retrofitting Analysis on Pre-trained GPT-2. Comparison of a vanilla pre-trained model against a VGA equipped pre-trained model, both with 600K iterations, and two fine-tuning strategies with 6K iterations. **Key Finding:** Fine-tuning with VGA, consuming only ~1% of the pretraining computation, significantly rectifies the pathology to a state comparable to scratch pre-training.

| Model | Method | Perplexity ($\downarrow$) | Max I+O Norm ($\downarrow$) | Avg. kurtosis ($\downarrow$) |
|---|---|---|---|---|
| | Vanilla | $17.03^{\pm 0.00}$ | $256.84^{\pm 18.77}$ | $13412.71^{\pm 3,528.30}$ |
| GPT-2 | VGA | $\mathbf{16.52}^{\pm 0.01}$ | $\mathbf{12.37}^{\pm 1.28}$ | $\mathbf{31.27}^{\pm 6.71}$ |
| | Fine-tuning w/o VGA | $17.01^{\pm 0.01}$ | $398.67^{\pm 43.88}$ | $9863.87^{\pm 2575.90}$ |
| | Fine-tuning w VGA | $16.94^{\pm 0.01}$ | $24.05^{\pm 4.73}$ | $32.47^{\pm 2.95}$ |

We investigated whether VGA could not only prevent extreme-token phenomena but also rectify them in an existing, pathologically-developed model—a concept known as retrofitting.

We performed a short fine-tuning run (6,000 steps) of a GPT-2 model on the data that was used in pre-training, comparing four variants: a vanilla pre-trained model, a VGA equipped pre-trained model, the model fine-tuned without modification (Baseline) and the model fine-tuned after retrofitting the VGA module (Retrofit).

The results in Table 3 decisively confirm VGA's retrospective correction capability. Standard fine-tuning (w/o VGA) failed to correct the pathology; in fact, the Max I+O Norm dramatically **worsened** from 256.84 to 398.67, confirming the self-amplifying nature of the optimization pathology when the VGA gate is absent. In stark contrast, retrofitting the VGA module immediately stabilized the model: Max I+O Norm plummeted from 256.84 to $\mathbf{24.05}$ and Kurtosis dropped to $\mathbf{32.47}$. This strong regularization yielded the best Perplexity in the fine-tuning group ($\mathbf{16.94}$). Crucially, these stabilization metrics effectively approach to those achieved by pre-training VGA from scratch, proving that VGA is highly effective even when introduced late in the learning process, significantly expanding its practical utility.

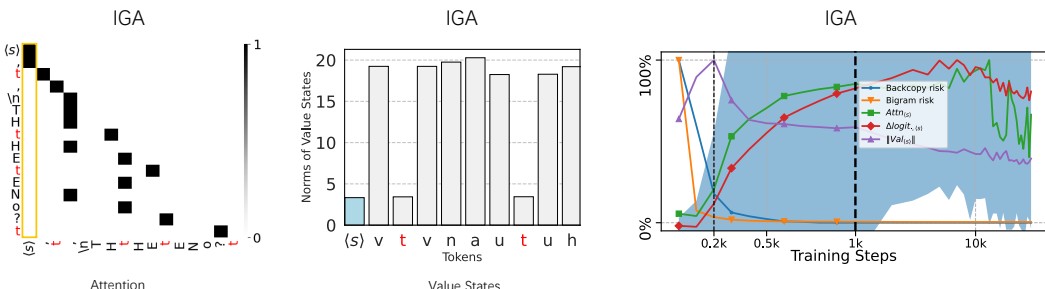

Figure 6: Analysis of IGA model on the Bigram-Backcopy task, left: Attention patterns, middle: Value status, right: Training dynamics.

# D ANALYSIS OF IGA MODEL ON THE BIGRAM-BACKCOPY TASK

As shown in Figure 6, IGA mechanism reveals a mitigation of the pathological dynamics observed in the Vanilla model, though its performance remains inferior to VGA.

In terms of Attention Sinks, while the Vanilla model exhibits a quintessential pathological collapse, where non-trigger query attention concentrates severely onto the  token (high $Attn_{}$), the IGA model successfully dampens this effect. IGA achieves a more distributed attention pattern than Vanilla, yet it does not eliminate the pathological concentration as comprehensively as VGA, which maintains $Attn_{}$ at healthy, non-extreme levels.

This partial alleviation directly translates into the Value-State Drains. The severe degradation of the start token's value-state norm, $||Val_{}|| \to 0$, which characterizes the Vanilla model, is mitigated in IGA, preventing an aggressive reduction towards zero. However, the $||Val_{}||$ in IGA does not stabilize at the maintained, non-pathological level achieved by the VGA model.

The distinct behavior is most clearly demonstrated in the Training Dynamics. The IGA model effectively slows and reduces the magnitude of the increase in the attention-related metrics ($||Val_{}||$ and $Attn_{}$), indicating a weaker pre-softmax bias and a less destructive feedback cycle compared to Vanilla. Nevertheless, the VGA model stands out by demonstrating fundamentally different training dynamics, where all pathological signals ($\Delta logit._{}$, $Attn_{}$ and $||Val_{}||$) remain stable throughout training, confirming that VGA is the most effective mechanism for breaking the mutual reinforcement cycle.

# E STATEMENT ON THE USE OF LLMS

During the preparation of this manuscript, we utilized large language models (LLMs) to assist with language editing and refinement. The purpose was to improve clarity, grammar, and overall readability. All scientific contributions, methodologies, and analyses are the original work of the authors, who take full responsibility for the content of this paper.

