# OpenReview forum: "Value-State Gated Attention for Mitigating Extreme-Token Phenomena in Transformers"
_ICLR.cc/2026/Conference — ICLR 2026 Conference Withdrawn Submission_

### Official Review · Reviewer_A7mK · 2025-10-28

**Soundness:** 2
**Presentation:** 2
**Contribution:** 2
**Rating:** 6
**Confidence:** 2

**Summary:**

This paper introduces a learnable gating mechanism driven by each token’s value vector to modulate its attention output, directly breaking the feedback loop behind extreme-token pathologies like attention sinks and value-state drains. This architectural fix markedly improves Transformer stability, model performance, and quantization fidelity.

**Strengths:**

1.VGA introduces a value-based gate that reactively regulates attention outputs, effectively breaking the feedback loop behind attention sinks—an improvement over prior input-gated methods.

2.The gradient analysis clearly shows how VGA decouples attention magnitude from value norm suppression, providing a principled fix to the mutual reinforcement cycle.

3.Tests on BERT, GPT-2, and OPT show VGA reduces activation outliers and improves stability without hurting perplexity, outperforming register tokens, learnable sinks, and IGA.

4.VGA yields exceptional INT8 post-training quantization robustness, with negligible performance loss compared to severe degradation in baselines.

**Weaknesses:**

1.Evaluations are limited to ~125M-parameter models; behavior on billion-scale LLMs remains unknown.

2.VGA requires architecture modification and retraining or fine-tuning, limiting plug-and-play adoption.

3.Experiments focus on language modeling only; generality across modalities or downstream tasks is unverified.

4.Slightly lower raw perplexity than some baselines (e.g., register tokens), suggesting it optimizes for stability over peak task accuracy.

**Questions:**

1.How does VGA scale to large-scale LLMs and long-context attention?

2.Can VGA be retrofitted into pretrained models via fine-tuning, or must it be trained from scratch?

3.Will value-based gating also help in non-language domains such as ViTs or multi-modal Transformers?

---

> ### Author Response · Authors · 2025-11-27
>
> **1. Interpretation of VGA Results and the Accuracy-Stability Trade-off.** We respectfully clarify the interpretation of our results in Table 1 and
> Table 2 to address the concern regarding a trade-off between stability
> and accuracy.
>
> - **1). Perplexity and Task Performance:** In language modeling, lower
> perplexity denotes superior performance. As shown in Table 1, VGA
> achieves lower (better) or equivalent perplexity compared to baselines
> in the majority of cases. Consequently, our results demonstrate that VGA
> enhances model stability without sacrificing peak task accuracy in the
> majority of cases.
>
> - **2). Quantization Robustness vs. Register Tokens:** While Register
> Tokens may yield competitive FP32 perplexity, they induce high Token
> Norm and Kurtosis. These outliers cause significant degradation during
> INT8 Post-Training Quantization. In contrast, VGA decouples value norms
> from attention scores. This architectural advantage allows VGA to
> maintain high accuracy in FP32 while simultaneously ensuring robustness
> in low-precision settings, offering a superior solution for
> deployment-ready models.
>
> **2. Using VGA for Fine-tuning.** We appreciate the suggestion, following which encouraging results are
> demonstrated, as discussed in the General Response to All Reviewers.

---

### Official Review · Reviewer_oSS8 · 2025-10-30

**Soundness:** 3
**Presentation:** 3
**Contribution:** 3
**Rating:** 6
**Confidence:** 3

**Summary:**

This paper introduces Value-State Gated Attention (VGA), an modification for Transformer models that aims to mitigate extreme-token phenomena (attention sinks, value-state drains). The authors provide an analysis of the gradient dynamics to motivate the mechanism of VGA, arguing VGA decouples high attention allocation from the destructive suppression of value norms. Experimental validation is conducted on a synthetic task and language modeling task, and quantization, show improve performance.

**Strengths:**

- The motivation is clear and is supported by an empirical validation on a controlled synthetic task.
- VGA shows some improvements on language modeling benchmarks, including better perplexity as well as quantization results.

**Weaknesses:**

- Reported results lack standard deviations or error bars, making it difficult to assess the reliability and statistical significance of the improvements.
- The paper claims VGA is a general enhancement applicable to any Transformer-based model, but it lacks evaluations beyond language tasks, such as in vision Transformers (e.g., ViT), which would strengthen the generalizability argument.
- An experiment illustrating the disadvantages of IGA over VGA would be valuable. For example, could you extend the results in Figures 4 and 5 to include IGA, showing how it fails to fully mitigate attention sinks or value drains in the same settings?
- Please provide more details on the creation of Figure 5. Explain the annotations (e.g., dots, dashed lines) and why specific training steps like 0.2k and 1k are marked?

**Questions:**

See weaknesses.

---

> ### Author Response · Authors · 2025-11-27
>
> ### **1. Error Bars and Figure 5.**
>
> - **Variance:** The shaded regions in **Figure 5** represent the
>   standard deviation across 3 training seeds. We have explicitly add
>   standard deviation values ($\pm \sigma$) to the main results (Table 1 & 2) in
>   the revised manuscript.
>
> - **Figure 5 Annotations:** The dashed vertical lines at 0.2k and 1k
>   steps mark the critical phases of sink formation. 0.2k corresponds to
>   the "early search" phase (finding a high-frequency token), and 1k is
>   where the sink behavior solidifies in Vanilla models. We will clarify
>   this in the caption.

---

### Official Review · Reviewer_PFsj · 2025-11-01

**Soundness:** 3
**Presentation:** 3
**Contribution:** 3
**Rating:** 6
**Confidence:** 4

**Summary:**

This paper aims to address extreme-token pathologies of transformers, including attention sinks and vanishing value-state vectors. By first attributing these phenomena to the mutual reinforcement loop in a typical softmax attention layer, the authors draw upon control theory to break this loop by proposing the Value-State Gate Attention (VGA) mechanism. In a nutshell, VGA adds a simple gating mechanism that is computed based on the value vectors V to regulate the gradient flow to them. The authors identify that such a mechanism enables a self-regulatory term that dynamically and adaptively adjusts the gradient path, effectively avoiding value drain. Experiments on a synthetic Bigram-Backcopy task and on BERT, OPT-125M, and GPT-2 (124M) show the effectiveness of the proposed mechanism.

**Strengths:**

1. The method is well-motivated and addresses the mutual reinforcement loop in a novel, mechanistic way.
2. The method is very simple, and can be incorporated into existing practices with minimal overhead.
3. The paper is also well-written and clear in presentation.
4. Experiments show consistent gains by this simple fix. A nice bonus is the promising results on low-precision settings.

**Weaknesses:**

1. This is a nitpick. While the experiments show promising results, it is a bit limiting in terms of scales as the experiments only considered sizes of ~100M. This is vastly smaller than modern models of billion-scale parameters. It is therefore a question of whether the same gains can be achieved on larger-scale models.
2. The experiments mainly focus on language modeling. I think assessing the effectiveness of the proposed method on more diverse task domains such as vision will greatly improve the work.
3. Forgetting gate mechanisms are widely used now and there have been many design choices: per-channel gates, temperature in sigmoid, normalized value states. The current design is simplistic, but could benefit greatly from ablating these different design choices to further enhance the performance of the method.

**Questions:**

Please see the weaknesses section.

---

> ### Author Response · Authors · 2025-11-27
>
> **Design Choices and Ablations.** We prioritized a minimal design to demonstrate that the core
> principle—negative feedback via Value states—is sufficient.
>
> - **Per-channel vs. Per-head:** As noted in our response to Reviewer
>   XV3o (Item 3), we prioritize our study on the source of the gating
>   signal rather than the design of the gating mechanism, adopting the
>   established IGA architecture. We hypothesize that per-head scalar
>   gating is sufficient because the attention pathology affects the
>   head’s output globally rather than on specific channels. While a
>   denser per-channel formulation increases parameter count ($d \times d$ vs. $d \times h$) and computational overhead, we empirically found no
>   commensurate performance gains to justify the added complexity.
>
> - **Normalization:** We intentionally omitted LayerNorm on the input to
>   the gate ($V$). The unnormalized magnitude of $V$ carries the critical
>   signal indicating whether a "drain" is occurring; normalizing it would
>   eliminate the very variance the gate uses to detect and suppress
>   outliers.

---

### Official Review · Reviewer_XV3o · 2025-11-01

**Soundness:** 3
**Presentation:** 3
**Contribution:** 3
**Rating:** 4
**Confidence:** 3

**Summary:**

The paper proposes Value-State Gated Attention (VGA), a lightweight and simple architectural add-on for Transformers that combats “extreme-token” pathologies - attention sinks and value-state drains. The key idea is a reactive, negative-feedback gate computed from the value vector $V_j$ itself, which multiplicatively modulates a token’s contribution at the attention head output. A gradient analysis argues this decouples pressure on value norms from attention-score updates, breaking the mutual-reinforcement loop that drives sinks/drains. Empirically, VGA reduces sink formation on a synthetic task and improves activation stability, perplexity, and post-training quantization (PTQ) robustness on BERT/OPT/GPT-2, with negligible overhead.

**Strengths:**

- The method is simple and clear. Turning the gate into a function of the value state (not the input) is a crisp design that directly targets the failure mode; the negative-feedback interpretation is compelling.

- The gradient pathway analysis makes the stabilization story plausible and distinguishes VGA from input-gated variants.

- Minimal code/param/compute overhead; orthogonal to attention-score computation; drop-in for many Transformer flavors.

- Presented experiments contain synthetic validation, standard LM backbones, and a relevant application, where extreme activations are especially harmful. Results consistently show fewer sinks, stabler value norms, and quantization gains.

**Weaknesses:**

- My main concern is that empirics are limited to a small set of baselines. Stronger comparisons against other sink-mitigation families (register tokens, softmax alternatives/clipping, predictive gates, state interventions) would better position VGA.

- No evidence at very large scales or on long-context regimes where sinks/drains become acute. It’s unclear how VGA interacts with KV caching, RoPE/positional schemes, and very deep stacks.

- There should be a formal metrics for the determination of “extreme tokens.” While qualitative/aggregate indicators are shown (norm stabilization, performance), clearer, standardized sink/drain metrics (incidence rates, attention concentration statistics, gradient norms) would strengthen claims.

**Questions:**

- Is $g_j$ is a scalar (as Eq. 6 suggests)? Any results with vector (per-dimension) gates or applying the gate before vs. after the output projection $W_O$?

- Are gates shared across heads or learned independently? Any empirical difference?

- Any preliminary results on vision or multi-modal Transformers where sink-like effects also appear?

---

> ### Author Response · Authors · 2025-11-27
>
> ### **1. Compatibility with Modern Architectures and Efficiency Optimizations.**
>
> We acknowledge the importance of integrating with modern architectural
> standards. In this section, we discuss the compatibility of VGA with
> RoPE and KV-Caching, and demonstrate its interoperability with Flash
> Attention to ensure efficiency and memory saving in practical training
> and deployments.
>
> - **RoPE:** As stated in our submission, we prioritized alignment with
>   modern architecture standards by integrating RoPE into all GPT-2
>   experiments, despite its absence in the default nanoGPT codebase. RoPE
>   and VGA are orthogonal and fully compatible: RoPE is applied to the
>   queries ($Q$) and keys ($K$), while VGA operates exclusively on the
>   values ($V$).
>
> - **KV-Caching:** To align with IGA, we opt to use a single channel gate
>   value per head. This value can either be cached alongside $V$ or
>   re-computed from the cached $V$, incurring only marginal overhead in
>   storage or computation, respectively. The optimal strategy should be
>   determined via profiling. Regardless of the choice, both methods
>   require only minor modifications to existing KV-Cache architectures.
>
> - **Flash Attention:** As noted, VGA is "orthogonal to attention-score
>   computation"—the specific process addressed by the Flash Attention
>   algorithm—rendering it fully compatible with standard Flash Attention
>   implementations. In contrast, the Learnable Sink operates directly
>   within the scaled dot-product attention mechanism. Consequently,
>   achieving Flash Attention-level performance with a Learnable Sink
>   would require a custom kernel design to support this additional
>   feature.
>
> ### **2. Formal Metrics for "Extreme Tokens" and Attention Sinks.**
>
> We appreciate the reviewer’s push for formalization. We utilize Token
> Norm and Kurtosis as the key metrics for studying the problem, because
> they are the direct mathematical definitions of the phenomena in
> question, consistent with established literature Bondarenko et al.
> (2023).
>
> **Token Norm is the definitive measure for "Extreme Tokens".** Since an
> "extreme token" is defined by having a feature magnitude significantly
> larger than the population mean, the norm directly quantifies this
> signal energy. It allows us to distinguish critical outliers from
> standard variance without introducing arbitrary thresholds.
>
> **Kurtosis is the standard statistic for "Attention Concentration".**
> Attention sinks create highly leptokurtic distributions (a single
> "spike" on the sink token against a flat background). Kurtosis is
> mathematically superior to variance or entropy for detecting this
> specific "winner-take-all" sparsity, acting as a robust fingerprint for
> sink/drain behaviors.
>
> Together, these metrics provide a complete and formal view: Norm
> measures the strength of the outlier, while Kurtosis measures the
> structural sparsity of the attention mechanism.
>
> ### **3. Details on Gating.**
>
> Our study prioritizes the source of the gating signal rather than the
> design of the gating mechanism. Since architectural variations were
> extensively explored in prior IGA work, we adopt their recommended
> architecture: one channel per head with independent weights. While the
> main text employs single-head notation for simplicity, full details
> regarding the multi-head implementation are provided in the Appendix. We
> will ensure this distinction is clarified in the revised manuscript.

---

### Author Response · Authors · 2025-11-27
**General Response to All Reviewers**

We thank the reviewers for their insightful feedback and for recognizing
the novelty and simplicity of our approach. We are encouraged that the
reviewers found our method to be *"simple and clear"* with a *"crisp
design"* (Reviewer XV3o), *"well-motivated"* and *"novel"* (Reviewer
PFsj), *"supported by an empirical validation on a controlled synthetic
task"*(Reviewer oSS8), and *"yields exceptional INT8 post-training
quantization robustness"* (Reviewer A7mK).

We are writing this general response to address the common requests for
billion-scale pre-training and extending the experiments into broader
domains raised by several reviewers.

**Regarding Large-Scale Experimentation and Domain Expansion.** We fully agree with the reviewers that validating our method on
larger-scale models and a broader model families would be scientifically
valuable to further establish its scalability and generality.

We sincerely apologize that we are unable to accommodate these requests
during the rebuttal period. We respectfully ask for your understanding
regarding the resource constraints we face. Access to massive-scale
compute is not uniform, even within the industry. Whether in academia,
research institutes, or corporate R&D divisions, compute budgets are
finite and strictly allocated. True "unlimited" resources are typically
reserved for a select few production-critical foundation models, whereas
research groups—regardless of affiliation—operate with budgets designed
for methodological exploration rather than capital-intensive
pre-training.

While we deeply appreciate the resource support that has been granted to
this study, it is non-trivial for us to request additional capacity
beyond our allocation. We estimate that even a single billion-scale
training run (e.g., 1.5B GPT-2 XL) would require approximately 26 8×H20
GPU-days on our available infrastructure. Consequently, the volume of
desired computation is unfortunately beyond our capacity.

**Our Focus: Depth and Rigorous Mechanism Isolation.** In lieu of pursuing breadth through massive scaling, we have focused our
rebuttal efforts on depth to ensure the robustness of our claims within
a feasible scope. We hope the reviewers can evaluate our contribution
based on the rigorous isolation of the algorithm’s properties we have
demonstrated. To address concerns about robustness and generalization,
we have conducted the following new experiments during this rebuttal
period:

- **Fine-tuning**: Following Reviewer A7mK’s suggestion, we investigated
whether VGA could not only prevent extreme-token phenomena but also
rectify them in existing models. We took a vanilla GPT-2 pre-trained
with 600K iterations and fine-tuned it with VGA for 6K iterations—1% of
those in pre-training. The results are shown below:
| Model | Method | Perplexity ($\downarrow$) | Max I+O Norm ($\downarrow$) | Avg. kurtosis ($\downarrow$) |
| :--- | :--- | :--- | :--- | :--- |
| **GPT-2** | Vanilla | $17.03^{\pm 0.00}$ | $256.84^{\pm 18.77}$ | $13412.71^{\pm 3,528.30}$ |
| | VGA | $16.52^{\pm 0.01}$ | $12.37^{\pm 1.28}$ | $31.27^{\pm 6.71}$ |
| | Fine-tuning w/o VGA | $17.01^{\pm 0.01}$ | $398.67^{\pm 43.88}$ | $9863.87^{\pm 2575.90}$ |
| | Fine-tuning w VGA | $16.94^{\pm 0.01}$ | $24.05^{\pm 4.73}$ | $32.47^{\pm 2.95}$ |

Fine-tuning with VGA successfully rectified model pathologies, reducing Max I+O Norm (256.84 → 24.05) and Avg. Kurtosis (13,412.71 → 32.47) while improving Perplexity (17.03 → 16.94). These metrics approximate those of models pre-trained from scratch, demonstrating VGA's efficacy in retrofitting existing models (details in Appendix C).

- **Illustrating IGA on BB-Task**: Following Reviewer oSS8’s suggestion,
we extended the experiments to include the IGA baseline. The results
visually demonstrate that IGA fails to fully mitigate value-state
drains, confirming that its input-based gating cannot effectively break
the mutual reinforcement cycle. The details are presented in Appendix D
of the revision.

- **Reliability and Statistical Significance**: Following Reviewer oSS8’s
suggestion, we conducted multiple independent runs with different random
seeds for all experiments, following the settings in Bondarenko et al.
(2023), and added standard deviations to the reported metrics. These
results confirm that **the performance gains and stability improvements
of VGA are statistically significant and robust**. The details are
updated into Table 1 and Table 2 in the main paper of the revision.

We believe these consolidated results, combined with our initial
validation across three distinct architectures (BERT, OPT, GPT-2),
provide strong evidence of the method’s efficacy and reliability. We
kindly ask that you evaluate the submission on its algorithmic novelty
and the comprehensive analysis provided, rather than penalizing it for a
lack of massive-scale compute resources.

We have addressed all other specific technical questions in our
individual responses below.

Sincerely,

The Authors

---

### Note · Authors · 2026-01-16

**Comment:**

Dear Area Chair and Reviewers,

We would like to express our sincere gratitude for your time and insightful feedback on our submission.

Following the rebuttal period, we secured additional computational resources to conduct more extensive experiments. During this process, we unfortunately and fortunately uncovered a bug in our experimental code. We wish to clarify that this bug is strictly isolated to the experiments involving the GPT-2 model; our other experimental results and their corresponding conclusions remain unaffected and valid.

However, given the importance of the GPT-2 results to the paper’s current narrative, we made the immediate decision to withdraw our submission upon discovering this issue.

We apologize for any inconvenience this causes and thank you again for your valuable contribution to the review process.

Sincerely,

The Authors

**Withdrawal Confirmation:**

I have read and agree with the venue's withdrawal policy on behalf of myself and my co-authors.